# Elevated Serum Levels of IgG4 in Patients with Heart Failure with Reduced Ejection Fraction: A Prospective Controlled Study

**DOI:** 10.3390/biology11081168

**Published:** 2022-08-04

**Authors:** Igor Volodarsky, Anamaria Anton, Liaz Zilberman, Irina Fugenfirov, Eran Neumark, Stephen Malnick, Yair Levy, Jacob George, Sorel Goland

**Affiliations:** 1The Heart Institute, Kaplan Medical Center, Hadassah Medical School, Hebrew University, Rehovot 76100, Israel; 2Department of Internal Medicine E, Meir Hospital, Kfar 4428164, Israel

**Keywords:** heart failure, ischemic cardiomyopathy, idiopathic dilated cardiomyopathy, IgG4 related disease

## Abstract

**Simple Summary:**

Immunoglobulin is a group of proteins participating in process of inflammation. One of its subtypes is Immunoglobulin Gamma-4 (IgG4) which is related to a rare disease called IgG4-related disease and characterized by a process of fibrosis—deposition of fibers in different organs and inflammation, resulting in tissue damage and dysfunction including the heart muscle and vessels. There is a paucity of information regarding the potential link between of IgG4 and cardiovascular diseases. The purpose of this study is to assess the serum levels of IgG4 in patients with dilated cardiomyopathy (DCM), a disease characterized by cardiac muscle dysfunction of ischemic and non-ischemic origin. Ischemic heart disease is a disease in which heart muscle is damaged by inappropriate blood supply caused by blood vessel obstruction and may cause heart muscle dysfunction. Ninety-eight patients with ischemic and non-ischemic cardiomyopathy were included in this study. The serum concentrations of IgG4 were measured. Patients with DCM had significantly higher levels of IgG4 compared with the healthy control group (77.4 ± 64.0 vs. 50.3 ± 28.8 mg/dL, *p* < 0.01). Although there was no difference in the total IgG levels in patients with ischemic DCM, the serum concentrations of IgG4 were significantly higher than the corresponding values in the control group (89.8 ± 67.3 vs. 50.3 ± 28.8 mg/dL, *p* < 0.01). In conclusion, we found that patients with ischemic cardiomyopathy had increased blood levels of IgG4. Additional studies with a larger number of patients are needed to support and explain this finding.

**Abstract:**

(1) Background: Immunoglobulin gamma subclass 4 (IgG4) is a serum protein belonging to the immunoglobulin superfamily. It has a central role in certain immune-mediated conditions defined as IgG4-related disease. There is a paucity of data regarding the potential association of IgG4 and cardiovascular diseases. Our aim is to study the serum levels of IgG4 in patients with ischemic and non-ischemic dilated cardiomyopathy (DCM). (2) Methods: patients with ischemic and non-ischemic DCM were included in this study. Non-ischemic DCM was defined as a left ventricular ejection fraction (LVEF) < 40% without coronary artery disease (CAD). Ischemic DCM was defined as a LVEF < 40% and proven CAD. The serum concentrations of IgG4 were measured by turbidimetry. (3) Results: Overall 98 patients with cardiomyopathy had significantly higher levels of IgG4 compared with the control group (77.4 ± 64.0 vs. 50.3 ± 28.8 mg/dL, *p* < 0.01). Although there was no difference in the total IgG levels in patients with ischemic DCM, the serum concentrations of IgG4 were significantly higher than the corresponding values in the control group (89.8 ± 67.3 vs. 50.3 ± 28.8 mg/dL; interquartile ranges: 40.4–126.5 vs. 31.8–66.8 mg/dL, *p* < 0.01). This was altered by gender and smoking. (4) Conclusions: The patients with ischemic DCM had increased serum concentrations of IgG4. Future studies are warranted to explore the potential role of an IgG4-mediated process in patients with heart failure with reduced LVEF.

## 1. Introduction

Dilated non-ischemic cardiomyopathy is one of the three traditional classes of cardiomyopathy, along with hypertrophic and restrictive cardiomyopathy [1]. The most frequent cause of dilated cardiomyopathy (DCM) is ischemic heart disease, and in this case, the disease is termed ischemic DCM. In other cases, other mechanisms are implied, such as an inherited disease, infections, and toxins [2]. Finding a specific cause for an individual case may be difficult, especially in patients with multiple risk factors. In most cases, however, the cause remains unexplained, and thus, these cases are designated as idiopathic. Some idiopathic cases may result from a failure to identify causes, such as an autoimmune etiology. The presence of left ventricular (LV) systolic dysfunction when other causes of dilated cardiomyopathy are ruled out is the hallmark of idiopathic dilated cardiomyopathy. Non-ischemic DCM remains a significant cause of systolic heart failure and the most common cause of heart failure in young people referred for cardiac transplantation [3]. With a prevalence estimated at 36–40 cases per 100,000 [1,4] and an incidence of seven new cases per 100,000 per year [1], this disorder affects well over 100,000 people in the United States alone. Generally, inflammatory pathogenesis is suspected [5]; however, an endomyocardial biopsy demonstrated inflammation in only a small subset of patients [6]. Treatment with immune-modulatory therapies has not proven effective for recent onset cardiomyopathy, likely due to the remote occurrence of the triggering event [7,8]. Cardiac biomarkers include a large number of enzymes, hormones, biologic substances, and other markers of cardiac stress and malfunction, and are extensively evaluated for treatment guidance and prognosis in non-ischemic DCM [9]. Many of them, including C-reactive peptide (CRP), amino-terminal pro–B-type natriuretic peptide (NT-proBNP), ST2 (also known as interleukin-1 soluble receptor), have been reported to provide different prognostic information in patients with ischemic and non-ischemic cardiomyopathy [9,10].

IgG4-related disease (IgG4-RD) is a recently recognized autoimmune disease with pathological features that are consistent across a wide range of organ systems. This entity may involve the pancreas, salivary glands, and many other organ systems [11]. An IgG4 molecule is presumed to play a central role in certain immune-mediated conditions. The diagnosis of IgG4-RD in the vast majority of patients is based on histopathologic features, such as the presence of IgG4 producing plasmacytes with elevated levels of serum IgG4.

There have been many reports of the potential involvement of IgG4-related immune activation in cardiovascular diseases, such as inflammatory abdominal aortic aneurysms, lymphoplasmacytic aortitis, and pericarditis [12,13,14]. Recently, rarer cases of cardiovascular involvements of IgG4-RD were reported in coronary periarteritis with aneurysms, myocardial ischemia [15,16,17], valvular lesions, heart blocks, and sudden death [18,19]. IgG4-related cardiovascular involvement may be incidentally diagnosed as a part of a multi-organ disease or as an isolated condition [20].

An increased serum concentration of IgG4 was found in patients with angiographically proven coronary artery disease (CAD), suggesting that the IgG4-related immuno-inflammatory process may play a role in the development and progression of coronary atherosclerosis, or alternatively, may mirror ongoing vessel wall inflammation [21]. The potential role of IgG4 in dilated cardiomyopathy has not yet been investigated.

In this study, we attempted, therefore, to explore the possible association between serum concentrations of IgG4 and non-ischemic dilated cardiomyopathy. As elevated levels of IgG4 were reported in patients with CAD, we investigated the serum concentrations of IgG4 in patients with ischemic cardiomyopathy.

## 2. Materials and Methods

### 2.1. Patients

Consecutive patients with heart failure with reduced left ventricular ejection fraction (HFrEFs) referred to our heart failure clinic between October 2010 and June 2013 comprised this prospective study. All participants signed informed consent forms. Patients with non-ischemic (those with a left ventricular ejection fraction (LVEF) < 40% in whom CAD was excluded by coronary angiography) and age-, LVEF-, and New-York Heart Association Functional Class (NYHA-FC)-matched patients with ischemic cardiomyopathy (patients with a LVEF < 40% in whom CAD was proven by coronary angiography) were included in this study. The demographic and echocardiographic data of the patients were collected and analyzed. Patients with primary valvular disease, mechanical valve prosthesis, unstable angina, recent myocardial infarction (within three months), and coexisting cardiomyopathy with a known etiology (e.g., toxic, etc.) were excluded. Patients with oncological diseases (or receiving chemotherapy), autoimmune or allergic diseases, chronic infections, and inflammatory processes were excluded. Patients in whom there was a suspicion that LVEF reduction was caused by a known clinical condition (e.g., tachyarrhythmia, use of alcohol, substance abuse, nutritional deficiencies, etc.) were excluded. Blood samples for CRP, ST2, NT-proBNP, and IgG4 levels were drawn. Since the normal ranges of IgG4 widely vary between populations, volunteers without known cardiac diseases were recruited, and their blood samples were drawn for levels of IgG4 to establish normal ranges of IgG4. This control group was compared with the study population by demographic data, comorbidities, and medications taken. The protocol conformed to the ethical guidelines of the Helsinki Declaration, and it was approved by our human research institutional review board.

### 2.2. Laboratory Measurements

Blood samples were obtained from all patients. The samples were then centrifuged within 15 min, and the serum was stored at −80 °C. The serum concentrations of IgG and IgG4 were determined by turbidimetry and an enzyme-linked immunosorbent assay (ELISA) (Invitrogen, Thermo Fisher Scientific©, Waltham, MA, USA, catalog number: BMS2095). The serum concentrations of NT-proBNP, the high-sensitivity C-reactive protein (hsCRP), and ST2 were measured by ELISA (Ray-Biotech©, Peachtree Corners, GA, USA, catalog numbers: ELH-proBNP-1, ELH-CRP-1, and ELH-IL1R4-1, respectively).

### 2.3. Statistical Methods

The mean, standard deviations, and interquartile ranges are presented, and a nonparametric Mann–Whitney test was performed to determine the statistical significance of the differences between the two groups. The original data were compared between the groups using nonparametric Kruskal–Willis tests. Wherever the Kolmogorov–Smirnov test showed a normal distribution, the data between the three groups were compared using a parametric t-test. *p* < 0.05 was considered significant. Statistical analysis was conducted using IBM SPSS version 21.0 (Armonk, NY, USA).

## 3. Results

### The Results of the Study

Ninety-eight patients with dilated cardiomyopathy admitted to our cardiac failure clinic were recruited. The baseline characteristics of the patients are presented in Table 1. Forty-four patients with ischemic cardiomyopathy (93% male, mean age: 57.5 ± 10.7 years) and 54 patients with non-ischemic or idiopathic cardiomyopathy (68% male, mean age: 57.1 ± 12.3 years) were included in this study. No significant differences were seen between the two groups in terms of the LVEFs (32.2 ± 6.1% and 31.9 ± 8.1%, *p* = 0.9 and NYHA FC, *p* = 0.7). Hypertension, diabetes mellitus, and hyperlipidemia were more prevalent among patients with ischemic cardiomyopathy (*p* = 0.02, *p* = 0.03, and *p* < 0.01, respectively). The patients developed ischemic cardiomyopathy at least three months after acute cardiac syndrome or coronary involvement. No patients had pancreatitis, salivary or lacrimal glands disease, tubulointerstitial kidney disease, thyroiditis, or idiopathic fibrosis in any organ system. Significantly higher levels of NT-proBNP, CRP, and ST2 were seen in patients with cardiomyopathy compared with the controls. Higher levels of ST2 were obtained in patients with ischemic cardiomyopathy compared with patients with idiopathic cardiomyopathy (717.1 ± 617 vs. 464.3 ± 352.5 pg/mL, *p* = 0.005). No difference was found in the levels of NT-proBNP and CRP between patients with ischemic and idiopathic cardiomyopathy (Appendix A). A weak correlation between the levels of IgG4 and ST2 was obtained (r = 0.2, *p* = 0.037) in patients with cardiomyopathy.

Forty-six volunteers without known heart diseases (43.8% male, mean age: 44.2 ± 8.6) were included in the study to define the normal levels of IgG4. Twenty-five percent of them had a diagnosis of hypertension, and 6% of them had hyperlipidemia treated with statins. The baseline characteristics of the controls are presented in Table 1, and they significantly differed between patients with ischemic and non-ischemic cardiomyopathy. The IgG4 level in the control group was 50.3 ± 28.8 mg/dL, which was considered a normal range. Overall, patients with cardiomyopathy had significantly higher levels of IgG4 compared with the control group (77.4 ± 64 vs. 50.3 ± 28.8 mg/dL, *p* < 0.01).

The levels of IgG4 in patients with ischemic cardiomyopathy, non-ischemic cardiomyopathy, and the control group are presented in Figure 1. The difference between patients with cardiomyopathy and the control group was mostly driven by the ischemic cardiomyopathy cohort showing significantly higher levels of IgG4 compared with the control group (*p* < 0.01). The levels of IgG in the three groups showed no significant difference (between ischemic and idiopathic cardiomyopathies: *p* = 0.18, ischemic cardiomyopathy vs. controls: *p* = 0.97, and idiopathic cardiomyopathy vs. controls: *p* = 0.23). When separately comparing the patients in different functional classes, no statistically significant differences were found in the levels of IgG4 (65.4 ± 53.9 vs. 94.6 ± 72.3 mg/dL, *p* = 0.08 for NYHA FC II and 75.4 ± 73.8 vs. 77.0 ± 57.7 mg/dL, *p* = 0.95 for NYHA FC III). When including gender and relevant comorbidities into the regression analysis, gender and smoking influenced the relationship between ischemic DCM and IgG4 levels (*p* < 0.01 and *p* = 0.03, respectively). The results are summarized in Table 2.

## 4. Discussion

### 4.1. Discussion and Interpretation of the Results

In this pilot study, we provided the first reported results of an increased concentration of IgG4 in patients with heart failure with reduced LVEFs, suggesting a link between elevated serum concentrations of IgG4 and ischemic dilated cardiomyopathy.

IgG4-related disease (IgG-RD) is a relatively novel autoimmune entity initially described in Japan in 2001 [22,23]. In the earliest publications, IgG4-RD was shown to manifest as autoimmune pancreatitis, but later, this condition appeared to involve a wide range of organs, including salivary and lacrimal glands [24], with certain histopathologic features. According to the most recent criteria, the diagnosis of the disease is based on three features: (1) histopathologic findings, (2) cytologic findings of plasmacytes in the inflammatory infiltrate, and (3) elevated levels of IgG4 in the serum [25,26]. Most of the authors agreed that these criteria were organ-specific. The ranges of elevated levels of IgG4 vary among patients with IgG4-RD. Although IgG4 concentrations decreased with steroid treatment, in many cases, they remained elevated [27]. Some studies showed that IgG4 levels were able to predict a relapse of the disease; however, most of those studies suffered from a lack of good follow-up [28]. Based on this information, elevated blood levels of IgG4 can be considered a possible indicator of the disease.

Among the four immunoglobulin G (IgG) subclasses (IgG1, IgG2, IgG3, and IgG4), circulating levels of IgG4 are the lowest (10–135 mg/dL) and account for less than 5% of the total IgG in healthy subjects. IgG4 is regarded as an anti-inflammatory protein and one of its roles is to competitively bind antigens that cause immune reactions and thus eliminate them from the system. This anti-inflammatory role of IgG4 led researchers to believe that IgG4 may not have been a causative factor, but it instead mirrored an ongoing or recent inflammation, while the primary causative factor remained to be defined [11].

In the last decade, several case reports describing the potential involvement of IgG4 in cardiovascular diseases were published. The most frequently reported cardiovascular presentations were aortitis (causing aortic aneurysms) [15,29,30,31] and pericarditis [32,33,34,35,36,37]. Subsequently, other forms of cardiovascular presentations were reported, such as valvular disease [18,38,39,40,41,42], heart block [18,43], and an involvement of heart chambers and coronary vessels with IgG4-related inflammatory pseudotumors [44,45,46,47]. Coronary artery involvement caused by coronary periarteritis with aneurysms and inflammatory pseudotumors surrounding and compressing the vessels [20,48,49,50,51] with subsequent stenosis or obstruction and ischemia have also been reported. In addition, cases of an acute coronary syndrome [15,16] and sudden cardiac death as the first presentations of IgG4-RD have been published [19,52,53]. Because fibrosis and lymphoplasmacytic infiltration are among the main hallmarks of this disease, the question arises about a possible association between IgG4-related disease and dilated cardiomyopathy.

The overexpression of several T-helper type-2-related cytokines, such as interleukins 4, 5, and 13 and anti-inflammatory cytokines, such as interleukin 10 and transforming growth factor-β (TGF-β) were demonstrated in IgG4-related immune reactions [28]. Most pathophysiological processes related to the different etiologies of dilated cardiomyopathy involve inflammation. Lymphocytic infiltrates, the activation of macrophages, T-cells and B-cells, and the elevation of an array of pro-inflammatory cytokines were all found during the clinical evolution of dilated cardiomyopathy [54]. Enhanced levels of TGF-β were found in patients with heart failure [55] and were also described in various animal models of cardiac remodeling and the transition from compensated hypertrophy to heart failure.

ST2, a soluble IL-1 receptor, is a biomarker reflecting pathophysiological processes, including inflammation and fibrosis. ST2 plays an important role in heart failure, represents a burden of congestion, and is especially related to left ventricular (LV) remodeling [56,57]. Increased ST2 levels in patients with symptomatic heart failure are associated with a worse prognosis [58,59,60,61]. In our patients, the levels of ST2 correlated with the levels of IgG4. The role of IgG4 in immune processes is well established and the existence of a correlation between ST2 and IgG4 in our patients with cardiomyopathy suggested that both these molecules may participate in interrelated processes involved in the pathophysiology of cardiomyopathy.

The findings of our study suggested that similar inflammatory mechanisms described in autoimmune diseases may play a role in the inflammatory response of myocardium to both ischemic injury and volume overload, leading to heart failure. Recently, Sakamoto et al. described increased IgG4 levels in patients with CAD without heart failure, suggesting that IgG-4-related immune system activation plays a role in the pathogenesis of coronary artery disease [21]. Taking into consideration the inflammatory nature of the myocardial response to ischemia and heart failure, there may be an elevation of IgG4 in heart failure, along with several other markers with a presumed anti-inflammatory role.

Atrial fibrillation (AF) was reported in IgG4-related disease [44]. However, in this study, there was no significant difference between ischemic and idiopathic DCM groups in terms of the diagnosis of AF, and numerically, there were more patients with AF in the group with idiopathic DCM than in the group with ischemic DCM. Between the entire cohort with DCM and the control group, no significant difference in the presence of AF was found. Therefore, the presence of AF cannot explain the differences in IgG4 levels between the cohorts.

A certain impact on the levels of IgG4 can be expected from the medications that our patients were taking. Angiotensin II, for example, has several roles in cardiac loading condition regulation, and it was shown to signal the production of pro-inflammatory cytokines [62]. The ACE inhibitor quinapril was shown to reduce experimental arthritis in mice [63]. This result may have been a group effect, and there is evidence that ACE inhibitors and angiotensin receptor blockers have anti-inflammatory properties by means of reducing angiotensin II levels or blocking its actions. To date, no correlation between the levels and actions of IgG4 and treatment with the inhibitors of the RAAS system has been observed.

A statistically significant difference in terms of IgG4 levels was found between the ischemic DCM and control groups, but not between non-ischemic DCM and control groups, suggesting that IgG4 may be mostly associated with inflammation as part of the atherosclerotic process and is less relevant in the case of the inflammatory process, leading to LV remodeling and myocardial fibrosis. Additional studies with larger number of patients will be able to elucidate this finding.

The results of this study suggested that IgG4-driven processes may cause inflammatory vascular injury, myocardial ischemia, and subsequently, ischemic cardiomyopathy. Considering these findings, we hypothesize that IgG4 plays a role in modifying the inflammatory process, leading to LV remodeling in patients with HFrEFs.

### 4.2. Study Limitations

The main limitation of this study was the lack of histopathology. Elevated serum levels of IgG4 may not have necessarily proven the causal role of IgG4 in LV remodeling and myocardial fibrosis. As we mentioned before, the results of our study suggested that IgG4 may play a role either in inflammation within atherosclerotic plaques or in inflammation leading to myocardial fibrosis. Thus, future studies on the immune histopathology of both myocardium and atherosclerotic plaques are needed. If a causal role of IgG4 in ischemic myocardial damage is established by histopathology, it may change the current management of patients with ischemic heart disease, as disease-modifying therapies may be helpful in preventing LV remodeling.

Another limitation of the study was the small sample size. However, being a hypothesis-generating study, its purpose was to raise the question of a possible link between an IgG4-related process and myocardial involvements in patients with HFrEFs by estimating the blood concentration of IgG4 in the patient population.

## 5. Conclusions

In this pilot study, we found elevated serum levels of IgG4 in patients with heart failure and impaired LV function, suggesting the possibility of an association between IgG4-related inflammation and dilated cardiomyopathy of an ischemic origin. However, this study failed to show a significant elevation of the levels of IgG4 in patients with non-ischemic dilated cardiomyopathy. Future studies including a larger number of patients and more homogenous groups are needed to explore the potential role of an IgG4-mediated process in patients with heart failure with reduced ejection fractions.

## Figures and Tables

**Figure 1 biology-11-01168-f001:**
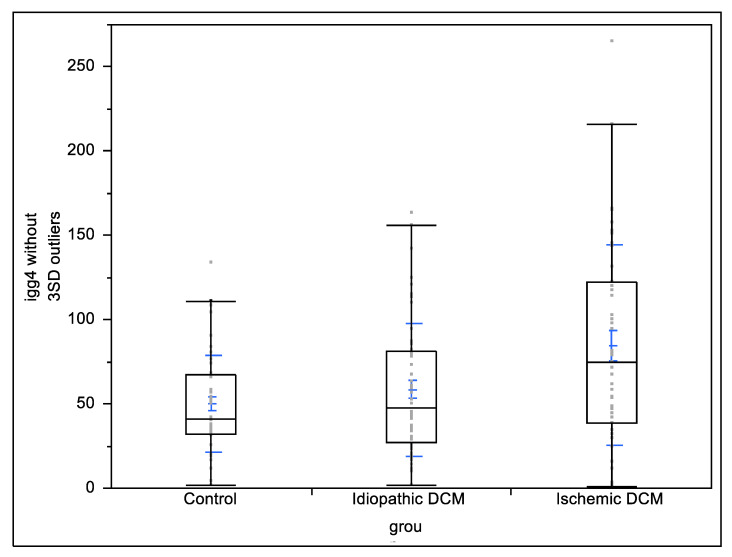
Comparison of IgG4 levels between patients with ischemic cardiomyopathy, non-ischemic cardiomyopathy, and control group. Black boxes represent interquartile ranges; blue bars represent mean and standard deviations.

**Table 1 biology-11-01168-t001:** Patients’ and controls’ baseline characteristics.

	Idiopathic DCM	Ischemic DCM	Control	*p*-Value Idiopathic DCM vs. Ischemic DCM
	N = 54 N (%)	N = 44 N (%)	N = 46 N (%)	
Age (years)	57.1 ± 12.3	57.5 ± 10.6	44.2 ± 8.6	0.9
Gender (male %)	34 (68%)	40 (93%)	20 (43.8%)	0.01
Diabetes mellitus	14 (26%)	21 (48%)	0 (0%)	0.03
Hypertension	32 (59%)	36 (82%)	11 (25%)	0.03
Dyslipidemia	37 (69%)	43 (98%)	3 (6%)	<0.01
Smoker	15 (28%)	21 (48%)	3 (6%)	0.06
Atrial fibrillation	6 (11%)	1 (2%)	0 (0%)	0.13
Implanted cardiodefibrillator	10 (19%)	13 (30%)	0 (0%)	0.23
Resynchronization therapy	16 (30%)	13 (30%)	0 (0%)	0.99
Diuretics	40 (74%)	40 (91%)	0 (0%)	0.04
ACEI	40 (74%)	28 (64%)	5 (11%)	0.3
ARB	8 (15%)	15 (34%)	6 (13%)	0.03
Digitalis	4 (7%)	3 (7%)	0 (0%)	1
Beta blockers	50 (93%)	43 (98%)	0 (0%)	0.4
LVEF (%)	33.1 ± 8.6	32.5 ± 7.0	N/A	0.7
NYHA FC				0.6
NYHA FC I	3 (5.6%)	1 (2.2%)	46 (100%)	0.6
NYHA FC II	33 (61.1%)	27 (62.4%)	0 (0%)	0.9
NYHA FC III	18 (33.3%)	16 (36.4%)	0 (0%)	0.9

Dilated cardiomyopathy (DCM), left ventricular ejection fraction (LVEF), New York Heart Association functional class (NYHA FC).

**Table 2 biology-11-01168-t002:** Comparison of total IgG and IgG4 subclass between patients with ischemic cardiomyopathy, non-ischemic cardiomyopathy, and control group.

	Overall DCM	Ischemic Cardiomyopathy (n = 44)	Non-Ischemic DCM (n = 54)	Control (n = 46)	*p* Value Ischemic vs. Non-Ischemic	*p* Value Ischemic vs. Controls	*p* Value Non-Ischemic vs. Controls
Total IgG (g/dL)	1.06 ± 0.78	1.19 ± 0.79	0.95 ± 0.58	1.1 ± 0.86	0.18	0.97	0.23
IgG4 subclass (mg/dL)	77.41 ± 63.69	89.77 ± 67.34	67.34 ± 59.94	50.28 ± 28.76	0.137	<0.01	0.37

DCM—dilated cardiomyopathy, IgG—immunoglobulin gamma, IgG4—immunoglobulin gamma subclass 4. The results are presented as mean ± standard deviation.

## Data Availability

Data generated and used to support results of this study are available and stored in our institutional database at Kaplan Medical Center.

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
