# Peer review of "Elevated Serum Levels of IgG4 in Patients with Heart Failure with Reduced Ejection Fraction: A Prospective Controlled Study"

_biology, 2022, doi:10.3390/biology11081168_

Round 1

Reviewer 1 Report

Dear Authors,

This is a very interesting and attention-grabbing manuscript.

I have only 2 notes to answer and correct:

  1. Allergic diseases were among the exclusion criteria, as they are known to increase IgG4 levels?
  2. 25% of healthy volunteers were not “healthy”, so I rather suggest to use the nomenclature “control group”.

I support the publication of the article.

Author Response

Thank you for your valuable advices. You can find the answers in the attached file

Reviewer 2 Report

Volodarsky, Goland and colleagues in their paper report some interesting findings from an analysis on the serum concentration of IgG4 in patients with ischemic or non-ischemic HFrEF compared to healthy controls. They found that the presence of LV systolic dysfunction is associated with higher subclass IgG4 concentrations. These results were also confirmed on multivariable analysis. The methods are sound. The field is interesting, despite at the moment for a main research aim.

I have the following concerns.

Major points

1) I am not convinced  about the DCM diagnosis. Indeed, only patients with LVEF <40% were included, while actually current recommendations for the diagnosis report that DCM diagnosis should be made in each patient with LVEF <40% (10.1093/eurheartj/ehv727). In my view, this is a cohort if ischemic and non-ischemic HFrEF.

2) IgG4 levels have also been associated with incidence of atrial fibrillation. In non-ischemic HFrEF, AF prevalence and incidence was proven to be an independent outcome predictor (doi: 10.1016/j.ijcard.2020.08.062.). May the IgG4 serum concentration be connected with this issue? Please discuss.

3) I see that control patients in some cases were on medical therapy (acei, arb) and had some risk factor (hypertension and diabetes). Please discuss if these comorbidities may influence the IgG4 concentrations.

Minor points

  • The manuscript would benefit from an English check.
  • Please report in the methods of the abstract that patients were compared with a healthy group population.
  • Please be more extensive regarding causes of DCM excluded (toxins, tachy-induced, hypertensive, chemotherapy)

Author Response

(The authors gave the same response as above.)

Reviewer 3 Report

To:

Editorial Board

Biology

Title: “Elevated serum levels of IgG4 in patients with heart failure with reduced ejection fraction. A prospective controlled study.”

Dear Editor,

I read this paper and I think that:

  • The authors should specify the number of involved patients in the abstract section. Please specify.
  • Furthermore, the authors should specify since the abstract what type of DCM they are writing about. Did they mean “idiopathic” DCM? Or secondary DCM? Or both? Please specify.
  • the small sample size is a limitation. Please discuss such a point in a dedicated limitation section.
  • A post-hoc sample size calculation should be performed.
  • All clinical characteristics of the study population should be described as they can impact on results.
  • Similar considerations are for comorbidities as they impact on results. The authors should better describe inclusion and exclusion criteria.
  • How many patients were on ICD/CRT therapy? This is an important issue to be addressed as it can impact on results.
  • The authors should perform a multivariate analysis in order to evaluate the role of confounding factors on final results. Please provide.
  • The burden of congestion in these patients should be addressed and discussed. The authors can consider and discuss the paper from Scicchitano P et al. Biomark Med. 2020 Feb;14(2):81-85.

Author Response

(The authors gave the same response as above.)

Reviewer 4 Report

This is generally a well-written and comprehensive article that reported increased concentration of IgG4 in patients with heart failure with reduced LVEF suggesting a link between elevated serum concentrations of IgG4 and ischemic dilated cardiomyopathy. The article is well-structured, very interesting and the results are presented in an appropriate manner, being clear and transparent. The statistical analysis is also well done. I consider that the study is valuable and sound and can be published in its current form. Even though this study included few patients and from a relatively small geographic area, I consider that the findings are interesting and that the results obtained can make significant contributions to further large studies. I want to ask the authors if they have thought about continuing the research on a large number of patients from different clinics?

Author Response

(The authors gave the same response as above.)

Round 2

Reviewer 3 Report

Authors well addressed my previous comments. the paper improved very much